# Women’s Health and Working Life: A Scoping Review

**DOI:** 10.3390/ijerph20021080

**Published:** 2023-01-07

**Authors:** Marianne Gjellestad, Kristin Haraldstad, Heidi Enehaug, Migle Helmersen

**Affiliations:** 1Department of Nutrition and Public Health, Faculty of Health and Sport Sciences, University of Agder, 4604 Kristiansand, Norway; 2Department of Health and Nursing, Faculty of Health and Sport Sciences, University of Agder, 4604 Kristiansand, Norway; 3Work Research Institute, Center for Welfare and Labour Research, Oslo Metropolitan University, 0176 Oslo, Norway

**Keywords:** female employees, occupational health, sickness absence, work ability

## Abstract

Women’s health matters for participation in working life. The objective of this study was to explore female physiology in a work–life context and to investigate possible associations between women’s health, sickness absence and work ability. A scoping review was conducted to develop a systematic overview of the current research and to identify knowledge gaps. The search strategy was developed through a population, concept and context (PCC) model, and three areas of women’s health were identified for investigation in the context of work. A total of 5798 articles were screened by title and abstract and 274 articles were screened by full text; 130 articles were included in the review. The material included research from 19 countries; the majority of the studies used quantitative methods. The results showed an impact on the occupational setting and an association between sickness absence, work ability and all three areas of women’s health, but a holistic and overall perspective on female biology in the work context is missing. This review calls for more knowledge on health and work and possible gender differences in this regard. Women’s health and working life involve a complex connection that has the potential to develop new knowledge.

## 1. Introduction

Work participation is linked to several socio-economic factors affecting living conditions and public health [1]. Thus, work–life is an important field for the equalisation of social differences and gender inequalities. It is also a crucial arena for reaching the UN’s sustainable development goals regarding health, equality and work [2]. However, both history and the present situation show a gender-divided labour market, especially regarding sickness absence [3,4,5,6]. Using Norway as an example, the complexity of gender segregation in working life is substantial. Even though Norway is one of the world’s most equally oriented countries, the labour market is still divided by gender; women work part-time more than men, and they take more sick leave [7]. Women also have greater representation in occupations with low status and lower representation in leading positions [7,8]. Norwegian working life has developed based on industrial workplaces since the last century [9]. Although the concept of health, safety and environment (HSE) has changed over the years, the Work Environment Act, written nearly 50 years ago, still consists of guidelines originally prepared for issues and challenges in elder, male-dominated work places [9,10]. Sickness absence rates are increasing for women (6.8%) as opposed to men (3.8%) [11], and Norway is at the top of the sickness absence statistics compared to other European countries [3]. The gender gap in sickness absence has been investigated in relation to different structural and cultural contexts [12]. Possible causes, such as gender differences in the burden of care for children and relatives, workload, nature of the work, work environment, career choices and attitudes, have been studied and resulted in various conclusions, though have not achieved a common understanding of the reasons for women’s higher rate of sick leave [4,7,12,13,14,15,16]. 

Women and men face different health challenges due to differences in biology and physiology [17,18]. This may influence women’s work participation and, thus, public health in general. Although reports and research from the last decade have proposed biological differences as a possible explanation for the gender difference in sickness absence [12,18,19,20,21], women’s health has not been sufficiently explored in the context of work. Women’s health can be defined as ‘diseases and ailments that only women have, diseases that affect more women than men, that affect very many women or that have different consequences for women than for men’ [22] (p. 27). Throughout the course of a normal lifespan, women undergo vast physiological and psychosomatic changes, both within the menstrual cycle and through the different stages of age [23,24,25]. These changes are normal processes, but they can also cause various imbalances and diseases. The monthly hormone cycle involves conditions and issues, such as premenstrual syndrome (PMS) and dysmenorrhea, which affect approximately 40–70% of fertile women [24,26,27] and endometriosis, which is prevalent in 10% of women [28]. The conditions may last throughout the woman’s fertile age until menopause. Perimenopausal symptoms, often experienced as heat, sweat, sleeping problems and depression, can cause discomfort for years before and after final menstruation [23,29]. Pregnancy was addressed in the context of Norwegian working life in the late 90s, with a focus on risk assessment and facilitation of the working environment [30,31], but pregnancy-related sick leave is still considered the major cause of the gender difference in sickness absence [32]. As many as 15% of women or couples also need medical assistance to get pregnant [23], and hormonal and other medical treatments for infertility can have severe side effects. In addition, abortions, both induced abortions and miscarriages, can burden women’s health [23]. Furthermore, diseases and conditions that mainly affect women are more often chronic and have non-specific symptoms [7,17]. An example of this is the experience of headaches; a current report shows a 43% prevalence of permanent or recurrent headaches in Norwegian women aged 25–44 years [7]. Female biology can be seen as more complex by nature than male biology due to both normal variations or changes and disposition for various diseases [7], which might influence work–life participation. Our objective is to explore the burden of female physiology in the context of work, with a broad approach to women’s health, and to investigate possible associations between women’s health, sickness absence and work ability. By developing a systematic overview of current international research, we will explore and summarise established knowledge and identify existing knowledge gaps. Objectives, research question, inclusion criteria and methods were documented in a protocol according to the theory of Peters et al. [33], for use within the research team. 

## 2. Materials and Methods 

We followed the five steps for scoping reviews described in the methodological framework of Arksey and O’Malley [34], together with the PRISMA guidelines and methodological considerations from the Joanna Briggs Institute [35].

### 2.1. Stage 1: Identifying the Research Question

Based on this objective, the following research questions were formulated:How has women’s health been investigated in the context of working life?What is the association between women’s health, sickness absence and work ability?

### 2.2. Stage 2: Identifying Relevant Studies

We selected studies based on predetermined inclusion and exclusion criteria. To obtain a broad approach to the field, we searched for studies using quantitative and qualitative methods. Only primary studies were included to access first-hand research experience. A time span of 10 years (2012–2022) was set to identify trends or developments over the last decade. Only studies presented in the English language were considered due to the international scope and limited time and resources at hand. Because work life is culture- and context-dependent, only studies from countries with transferability to the Norwegian context were considered, namely European countries, the USA, Canada, Australia and New Zealand. The search strategy was developed using a population, concept and context (PCC) model [36]. 

#### 2.2.1. Population

The population was women/females employed in paid work. No age limit was set for the population, as we wanted to include women at all stages of occupational life. Due to this objective, this study concerned women with female biology regarding hormones and sex organs. Thus, the extended gender-identity debate was not addressed.

#### 2.2.2. Concept

The term women’s health was conceptualised into three areas relevant for investigation in the context of work: (1) Life stages of hormonal impact (menstruation, PMS, dysmenorrhea, endometriosis and menopause), (2) Pregnancy and reproduction (pregnancy, breastfeeding, infertility and abortion), (3) Chronic and complex disorders (chronic pain, headache, dizziness, fatigue and fibromyalgia). The selection and formulation of these areas were based on previous research and literature on medicine and public health [17,22,37,38]. Conditions and diagnoses were included to make a relevant and sufficiently comprehensive selection and did not constitute an exhaustive list. Some of the areas were expected to complement each other and overlap with regard to symptoms. Key words were set up for the search using the Emtree/MeSH browser. 

#### 2.2.3. Context 

The context for the study was working life and the occupational setting. Outcome measures were sick leave/absenteeism, work ability, participation, presenteeism/reduced capacity and disability.

Literature searches were performed in the electronic databases MEDLINE/Embase (Ovid) and Scopus on 4 January 2022. See Table 1 for the search string. The search process was assisted by an experienced university librarian. 

### 2.3. Stage 3: Study Selection

The criteria for inclusion/exclusion are presented in Table 2. Due to the openness of the research questions and the breadth of the subject terms, the search identified many irrelevant articles, systematic reviews, and grey literature. These were excluded from our scope but nevertheless provided broader understanding and background knowledge.

A total of 5798 unique articles were screened by title and abstract for relevance to the inclusion and exclusion criteria, and 274 articles were screened by full text. Identified records were imported to EndNote and remaining duplicates were deleted. A minimum of two authors performed the same screening processes separately as the first author (M.G.) screened all articles and the co-authors (M.H., K.H., H.E.) each screened a third of the material. The electronic tool Rayyan was used for both steps. Disagreements around identified conflicts (max 15%, both stages) were discussed and resolved by all the authors. Finally, 130 articles were included in the review. The identification of studies is illustrated in Figure 1.

### 2.4. Stage 4: Charting the Data

Included articles’ author names, titles and publication years were transferred to Excel for charting and analysis. Articles were categorised by country, research design and the three defined areas of women’s health, including underlying conditions and measures on outcomes of working life. The aims and important results from each article were summarised in a separate column. 

### 2.5. Stage 5: Collating, Summarising, and Reporting the Results

A template for categorisation was created using the predefined concepts, and a consistent approach was sought through the systematic collating of the data. Categorisation was done by the first author (M.G.) and was discussed among the co-authors. Samples were taken in all categories to check consistency. 

## 3. Results

The 130 included articles were geographically spread among 19 countries. Overall, 26 articles had a qualitative design, 2 used mixed methods and 102 were quantitative studies. The countries and designs are shown in Figure 2. Together, the research in this review included data from 5,357,450 individuals.

Table 3 shows the distribution of the three main areas of women’s health, sickness absence and work ability. The results showed that 70articles were about pregnancy and reproduction, whereas life stages of hormonal impact were addressed in 39 articles and chronic and complex diseases in 21. The division into three main areas of women’s health was the basis for charting the data, but the conditions and symptoms for each area overlapped in the articles. 

Sickness absence was discussed in 77 articles, and work ability was discussed in 55 articles. The risk of illness was addressed in 30; the majority of these were about pregnancy and the work environment. Other measures used in the categorisation were presenteeism/reduced capacity, disability and participation, with 22, 15 and 18 articles, respectively. White-collar professions were investigated in 36 articles, whereas the rest included both white collar and blue collar or did not specify. No studies in the search specifically addressed blue-collar workers. Of the articles that specified professions, 33 involved nursing or health care, 10 involved teaching, 4 involved service, 2 involved industry and 17 involved other professions. Of the included studies, 33 were from the public sector. The rest were either from both the private or public sectors or did not specify. In 87 articles, professions were not specified.

### 3.1. Life Stages of Hormonal Impact

Women’s hormonal changes were found to impact work participation in most of the included studies of life stages. Menstruation and menstrual problems, including endometriosis, were addressed in 20 articles. The various conditions overlapped to a large extent in the different studies (e.g., menstrual problems described as symptoms of endometriosis). Both PMS and menstruation in general, as problems in the work setting, have been investigated in our material, with diverging results. Only two of the included studies [39,40] used qualitative methods to explore women’s experiences in this regard. Hardy and Hardie [39] described how PMS symptoms, such as concentration problems, fatigue, tearfulness and heightened sensitivity to both people and the work environment, might contribute to presenteeism and absenteeism. Sang et al. [40] argued that the menstrual cycle is ‘a problematic source of gendered inequalities at work’ (p. 1) and suggested supportive workplaces and public health policies to approach the problem. They identified several themes in their study, including managing pain, menstrual leaking, stigma and the importance of access to facilities. The burden of menstruation comprises difficulties that affect female participation, especially regarding reduced capacity and work ability [40]. The prevalence and severity of the symptoms are also confirmed by quantitative studies [41,42,43]. However, two quantitative articles from the USA concluded that menstruation explains the gender gap in sickness absence to a very little extent [44] or not at all [45]. Endometriosis was studied in 13 articles, of which 10 were quantitative studies and 1 used a mixed method, showing prevalence and significant risk for absenteeism, presenteeism and reduced work ability [43,46,47,48,49,50,51,52,53,54,55]. The included studies showed that endometriosis can affect the quality of daily working life but is also associated with impairment of professional life from a broader perspective, affecting career choices [55]. A study exploring the role of COVID-19 in women’s experiences [49] found that a home office could give women with endometriosis more flexibility, making it easier to take small breaks, avoid sickness absences and not have to hide associated pain. Menstruation was specially investigated in the context of shift work among nurses, showing diverging results: Lawson et al. [56] found that night work, working long hours and physically demanding work can lead to menstrual disturbances, while Moen et al. [57] found no association between menstrual characteristics and work parameters. Many of the studies addressed women’s health within several different contexts, with occupational life as only one area and did not specify the type of work. 

Menopause and work were investigated in 19 articles, 15 of which were quantitative. Menopausal symptoms, such as hot flushes, poor concentration, poor memory, sleep disorders, tiredness and feeling depressed, can affect work ability negatively [58,59]. One of the included studies showed that three-quarters of menopausal women with symptoms reported serious problems in meeting the demands of their work [60], and another study showed that many women face difficulties being open to managers and others about their problems [61]. A supportive environment with understanding employers and colleagues is important for women handling symptoms while at work [58,61,62,63]. Having an organisational policy for menopause is recommended [64], but it is also suggested not to isolate menopausal symptoms but rather to approach health and ageing more holistically [62,65,66]. Research on menopause has measured, to a large extent, work ability rather than sickness absence, but findings from the included studies show that menopause clearly impacts both [60,67,68,69,70,71,72]. Menopause can also be an economic burden because of absence from work and thus has a social impact on the women affected [71,73].

### 3.2. Pregnancy and Reproduction

Pregnancy and work were investigated in 56 articles. Of these, 39 addressed sickness absence, 11 work ability and 23 different risks for sickness in the mother or damage to the foetus. Fourteen of the studies were from the healthcare sector. The included studies also showed that pregnant employees are a group that needs facilitation and individual adjustments to maintain work ability and avoid sickness absence [74,75,76,77], and a positive and supportive work environment is crucial for pregnant women staying at work [78]. Other identified factors are a common understanding between the employer and the employee and sufficient resources in the organisation to make necessary adjustments for specific work tasks. The leader’s role is decisive, and the individual needs of the pregnant employee must be the main focus of measures and practical adjustments [79]. The high sickness absence rates for pregnant women are substantial and have complex causes [80]. Physically demanding work and physical job stressors in pregnant workers increase the risk of different outcomes, such as pelvic pain [81], miscarriage or preterm birth [82,83,84,85]. Some of the included studies showed that working nights and long shifts might increase the burden on pregnant employees. A study of Danish hospital employees found an association between night shifts and post-partum depression [86], and increased risk of hypertensive disorders in pregnancy [87]. Multiple occupational risks and exposures increase the use of sick leave during pregnancy [88,89]. Despite the high rates of sick leave in pregnancy, women report struggling with credibility when requests for sick leave or medical care are rejected [90]. This problem is addressed in studies on two common complaints—pelvic pain [90] and nausea [91]—which reported that women feel trivialised or not taken seriously when seeking guidance from health care professionals. One Norwegian study explored sickness absences in pregnant immigrant women and found that more immigrant women reported absences from work than native women. Possible causes were poorer health status prior to pregnancy, more severe pregnancy-induced emesis among immigrants and poor proficiency in the Norwegian language [92]. The risk for sickness absence in pregnancy increases when the woman is multiparous or overweight/obese, and is lower if the woman is engaged in leisure-time physical activities [93]. The risk of sickness absence is also linked to the level of education, as higher education is associated with less sickness absence in pregnancy. Younger mothers have greater sickness absences, possibly partly because young women are more likely to have a lower education [94]. Postponement of the first pregnancy does not explain the increase in pregnant women’s sickness absence [80]. Pregnancy does not predict greater sickness absence later in life; studies show that women who have previously given birth have less sickness-related absences and less use of disability pensions than women who have not given birth [95,96,97]. 

Breastfeeding was investigated in 12 articles, 9 of which were from the USA. Most countries in the Western world have national policies that guarantee breaks for breastfeeding for more than six months [98,99] and breastfeeding at work is a matter of organisational culture [100]. Scott [101] argued that experiences depend on individual, interpersonal and organisational factors and that policies are helpful for handling different needs in different roles. Employers’ accommodations are decisive for what is done in practice [76,102,103], and it seems easier for employers to facilitate pumping than breastfeeding [104]. A qualitative study from an Australian university setting identified a positive and progressive environment and private and safe spaces for breastfeeding as important factors for the women [105]. The same authors found that breastfeeding women might feel self-conscious and unprofessional and develop a resilience to judgement. Another finding of this study was that employees with permanent positions had better terms than temporary employees, and the authors suggested that the gender perspective is not sufficiently addressed in this regard [105]. 

No studies on abortion and the context of work were identified in our search. Infertility and work were addressed in four articles, all quantitative. Infertility treatment was shown to have a clear impact on work ability and sickness absence [53,106], and the overall professional impact of infertility treatment was significantly higher for women than for men [106]. In one study, 49% of respondents reported a negative influence on work ability and 46% reported the necessity to lie about missing work because of treatment. Another study found that the majority disclosed the treatment to their employer because of the need to take time off work [107]. 

### 3.3. Chronic and Complex Disorders

Of the 21 studies on chronic and complex disorders, 14 were studies on chronic pain. Chronic pain as a symptom is a great burden and has a clear impact on work ability and sickness absence for women suffering from disorders such as fibromyalgia [108,109,110] and endometriosis [48,50,51]. Chronic low back pain is found to be a possible consequence of particularly strenuous work involving walking and heavy lifting for women [111]. A study that investigated immigrant women and their experiences of being on sick leave due to chronic pain found this group particularly vulnerable to social isolation and dropout from working life [112]. Another study showed that self-reported recurrent headaches were associated with impaired productivity at work; the association with reduced work ability and presenteeism was clearer than the association with sickness absence or absenteeism [113]. In a study of Bulgarian nurses, 40% of the respondents reported frequent headaches [114]. No studies of dizziness in the context of work were identified in our search. Fatigue was addressed as a condition or symptom in studies about fibromyalgia and endometriosis, and two studies investigated shift work and work stress among nurses, with fatigue among the measured outcomes. Fatigue as a symptom can negatively impact work ability and increase sickness absence, both in women with fibromyalgia [115,116] and endometriosis [53,54,55]. Working night shifts was associated with chronic fatigue [117], and fatigue was also prevalent in more than 70% of the nurses in the Bulgarian study investigating work stress and long working hours [114]. The search identified three relevant articles on chronic conditions and work regarding female urogenital health. Two were from the health care sector, concluding that urine incontinence is a condition of high prevalence and significant severity in female nurses and midwives [118,119]. The experience of symptoms is associated with delayed voiding because of the organisation of the work, focus on patients at the expense of self-care, relationships in the nursing team, demands of the nursing role and inadequacy of workplace amenities [119]. Arcas et al. [120] argued that women’s absences from work are of longer duration than men’s and must be seen in association with family–work role conflict, calling for a holistic approach to the field. 

## 4. Discussion

This scoping review had two aims: to map the current knowledge on women’s health in the context of working life and to investigate associations between women’s health, sickness absence and work ability. We included and extracted data from 130 articles and found that some of the conditions within the field of women’s health are well-studied in the context of work, while others are weak or absent in the body of knowledge. Although individual studies have shown an association between sickness absence and work ability, a more holistic and overall perspective on the female burden in this context seems to be missing. 

The included studies showed that the hormonal system can cause symptoms that influence work participation [39,41,43,58,59]. Menstruation affects most women, and almost half of them experience pain or physical or psychological tension before or during the period, which can amount to a significant burden at the work place [24,27]. However, this scoping review showed diverging results. Menstruation in general was investigated and found to have weak associations with work participation, but other findings show that specific conditions and ailments have an influence [27,41]. Endometriosis, affecting 10% of women, together with menopause, affecting most middle-aged women and causing symptoms in 80% [29] represents a burden for a substantial number of female employees. In addition, the unclear distinction of whether symptoms stem from normal conditions or diseases can delay openness to the employer. The overall proportion of women negatively affected by hormonal conditions is still unknown, as the total prevalence includes overlapping conditions that have not yet been investigated. Thus, knowledge of its overall impact on women’s working lives is fragmented and limited. However, several of the identified articles suggest a more holistic approach to age-specific female ailments and emphasise that experiences are individual [65,66]. Research on life-stage policies should be investigated and promoted in this context.

Many of the included studies focused on pregnancy. Being pregnant is not a disease, but it can cause and involve conditions that hinder women from functioning normally at work, depending on their work tasks. National work–life legislation protects women against negative health consequences, but implementation of the laws will depend on employers’ resources and possibilities. There are also substantial differences between legislation in the various countries, which can have consequences for sickness absence and dropout from the workforce. This, however, is outside the scope of this review and should be addressed elsewhere. Pregnant workers in the healthcare sector have been studied specifically. In this sector, resources are scarce and managers have a broad span of control, so measures can be difficult to implement in practice [80]. Hospitals were also the study sites for several of the studies on breastfeeding in the workplace. The findings show a system under time pressure [101]. Experiences of adjustments are influenced by individual, interpersonal and organisational factors, and leaders are responsible both for developing policies and for making practical arrangements [101,121]. 

One of the studies on infertility found a severe impact on the working life of women undergoing treatment [106]. Considering how many women this applies to, it may represent a substantial burden for working women of fertile age. In many countries, infertility treatment is not a justified reason for sick leave. It is possible that the ailments are camouflaged in alternative diagnoses and one can question whether this affects stigma or lack of openness around the situation, but this is still unknown. Surprisingly, our results show that no studies on abortion and the context of work were identified by our search, so the consequences of abortion on women’s work participation seem to be unexplored. In particular, recurrent miscarriages can be assumed to have an impact, but this also represents a gap in knowledge. 

The prevalence of headaches/migraines and their association with work was the main objective of only one article in our search [113], although almost half of all women reported permanent or recurrent headaches in a recent report [7]. Research on chronic pain and fatigue in relation to work life also represents gaps in the knowledge, as most studies on this topic were general, with fatigue being only one of several measures. Evidence lacks both in quantity, since the number of studies is small, but also regarding the variety of research designs, as different approaches contribute different types of knowledge. 

### 4.1. Sickness Absence and Work Ability

Sickness absence and work ability are frequently used parameters in research on working life and occupational health. However, we found the terms used in various ways. Sickness absence is a measurable quantity that describes the time of absence from work. Still, the value depends on how it is measured. In some studies, data refer to sickness absence mandated by a doctor, partly because this is what generates numbers in national statistics [93,94]. Self-reported short-term absences are not part of this statistic and may be covered by the employer, depending on governmental policies. One can also ask how frequently short-term absence is associated with long-term absence. Some studies are based on self-reported absences, so the variations complicate comparisons and overviews. The term ‘work ability’ is also used in different ways. Work ability can refer to measurement by the Work Ability Index, a tool for assessing self-rated work ability, as used by Humeniuk et al. [59]. In other articles, it is used as a general term describing work ability or work capacity, such as by Hickey et al. [62]. The terms absenteeism and presenteeism for sickness absence and reduced work ability seem to be incorporated into the research on occupational health. The association between sickness absence and work ability was not explicitly addressed in our material, but the terms intertwine and overlap in different ways. 

Conditions from all three of the identified areas of women’s health—life stages of hormonal impact, pregnancy and reproduction and chronic and complex disorders—were found to influence sickness absence and work ability. Pregnancy contributes to a large part of the gender difference statistically, and despite knowledge of important factors for keeping pregnant women at work [78,79], high levels of sickness absence persist. Therefore, the findings of this review suggest a need for new research questions. If it is an aim to reduce sickness absence for pregnant women, studies of the arrangement of welfare benefits, workplace accommodations or how women can be sufficiently empowered to maintain health and support while working during pregnancy should be increased. 

Chronic conditions also show an association between work ability and sickness absence. Headache as a symptom was found to be more connected to presenteeism than absenteeism [113], which means that a huge proportion of women go to work despite having pain. Working while in pain may reduce work ability and thus challenge the experience of coping with and mastering an occupational setting. This, again, can lead to sickness absence over time. 

### 4.2. Need for Diverse Research

Most of the included studies provided quantitative knowledge of the prevalence and measurement of risks. To explore mechanisms more elaborately, studies using qualitative methods would contribute to a more detailed picture. The included qualitative studies mainly explored women’s experiences. Extension of this perspective would identify areas in which more knowledge is needed, as well as strengthen the basis for developing adequate measures and solutions for working women in practice. Research from the employer’s perspective is also scarce and would contribute important information. The high number of studies not specifying sector or profession favours general insights and obscures details. General and cross-sector research provides a potential basis for making policies and developing guidelines for HSE work. However, adjustments and measures for practice must be rooted in contextual and specific needs and possibilities in different professions. Division by gender was a criterion for inclusion/exclusion in our review, so all the articles had gender-specific findings. Still, few studies have investigated the rationale behind gender differences or their experiences. More knowledge of the identified differences will provide directions for future research on gender and health in occupational settings. 

## 5. Conclusions

Female biology makes women more disposed of different ailments that men naturally avoid. Some of these conditions are natural processes that cause bothersome symptoms, and some conditions are caused by illness. In this review, we wanted to investigate if and how female biology is associated with female work participation. Hormonal changes and menstruation-related ailments, pregnancy and reproductive matters and chronic and complex disorders were all found to impact the occupational setting. The existing knowledge is fragmented, as some of the conditions are investigated and others are not, and a holistic and overall perspective on female health in the context of work is missing. 

The search was done in only three databases, which leaves us at risk of missing important studies. Several conditions and diagnoses within women’s health were excluded from the search due to space limitations; this also represents a weakness of this study. However, we found the three defined areas relevant and comprehensive. When considering the studies on the various ailments, it becomes apparent that women’s total burden can challenge work participation. At the same time, participation in work life is health promoting by itself [1]. Therefore, coping in the workplace despite health challenges should be an important strategy for employers and policymakers. The importance of individual adjustments and organisational strategies for facilitation was also a finding for several of the conditions of the included studies [63,79]. This review calls for more knowledge on the associations and connections between health and work and the possible differences between women and men in these interactions. Many of the conditions and diagnoses discussed have a diffuse character, and some do not qualify for sick leave and are therefore not registered in the sickness absence registers. Hence, it is difficult to obtain a scientific perspective on women’s health issues related to work participation. Over time, living and working with ailments can cause wear and tear. In combination with difficult working conditions, this might lead to reduced work ability and high levels of sickness absence. Women’s health and working life constitute a complex connection with the potential for the development of new knowledge, and the scope of this connection should be considered from a broader perspective of living conditions and public health. 

## Figures and Tables

**Figure 1 ijerph-20-01080-f001:**
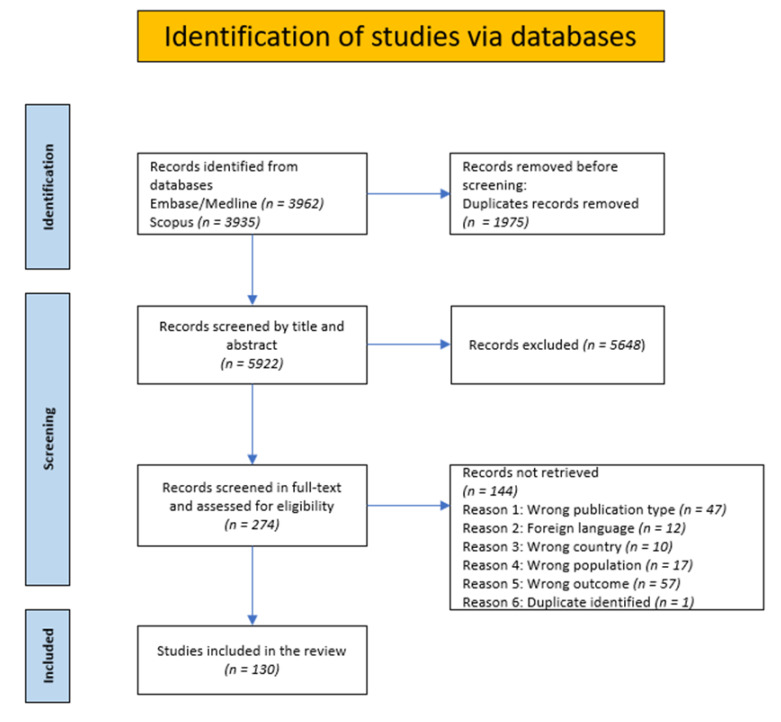
Flowchart of study selection.

**Figure 2 ijerph-20-01080-f002:**
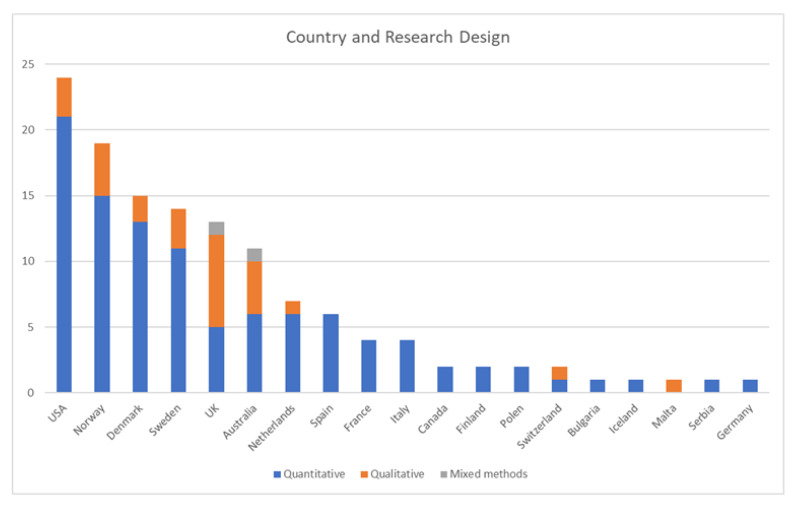
Bar chart of countries and research design.

**Table 1 ijerph-20-01080-t001:** Search string, database Embase/MEDLINE (Ovid).

	Query	Results from 4 Jan 2022
1	(women or women or female*).ti,ab,kf. or gender.ti. or female/	19,995,529
2	((professional* or work*) adj5 (life or ability or capacity* or participat*)).ti. or occupational health.ti,ab. or working health.ti,ab. or occupational health/	116,648
3	Sick Leave/ or presenteeism/ or Absenteeism/	38,414
4	(disability leave or illness day* or Absenteeism* or presenteeism* or sickness*).ti,ab.	68,304
5	(absence* adj3 (work or job)).ti,ab.	5394
6	(sick* adj3 (leave or day* or absence* or presenteeims*)).ti,ab,kf.	20,141
7	or/2–6	209,686
8	(Women’s Health or Maternal Health).ti,ab,hw.	115,652
9	(endometriosis or endometrioma or adenomyos*).ti,ab.	67,960
10	exp Menopause/ or (menopause or postmenopause or premenopause).ti,ab.	138,658
11	exp Menstruation Disturbances/ or Dysmenorrhea/ or (dysmenorrhe* or menstruation* or premenstrual or PMS).ti,ab.	128,981
12	exp genital system disease/ or (genital* adj3 disease*).ti,ab,hw,kf.	1,216,413
13	exp Genital Diseases, Female/ or exp “Female Urogenital Diseases and Pregnancy Complications”/ or pregnancy disorder/ or urogenital tract disease/	4,208,769
14	exp pregnancy/ or (pregnancy or pregnant).ti,ab. or (childbirth* or child birth* or breastfeeding or breast feeding or abortion or infertilit*).ti,ab,kf,hw.	2,232,523
15	(chronic adj4 pain).ti,ab. or Chronic Pain/	202,377
16	Fibromyalgi*.ti,ab,hw.	36,819
17	(Headache or head ache or Migraine or dizziness).ti,ab,hw.	512,936
18	((chronic adj3 fatigue) or (Myalgic adj2 Encephalomyelitis)).ti,ab. or chronic fatigue syndrome/ or Fatigue Syndrome, Chronic/	24,777
19	or/8–18	6,540,001
20	1 and 7 and 19	11,217
21	limit 20 to yr=“2012-Current”	5371
22	remove duplicates from 21	3962

**Table 2 ijerph-20-01080-t002:** Criteria for inclusion/exclusion.

Inclusion Criteria		
Methods	Primary studies of all types of methods	
Countries	Europe USA Canada Australia/NZ	
Language	English	
Population (P)	Females Employees (paid work)	
Health as concept (C)	Areas within the field of woman’s health: Life stages of hormonal impact (PMS, menstruation, dysmenorrhea, endometriosis, menopause) Pregnancy and reproductive health (pregnancy, breastfeeding, infertility, abortion) Chronic and complex disorders (chronic pain, headache/migraine, dizziness, fatigue syndrome/myalgic encephalomyelitis(ME), fibromyalgia)	
Measures and outcomes in working life (C)	Participation in working life Sickness absence Work ability Interventions on customisation/facilitation of work organisation (adjustments)	
**Exclusion criteria**		**Labelled reason for exclusion**
Methods	Systematic reviews Conference papers Grey literature	Wrong publication type
Countries	All other than the included	Wrong country
Language	All other than the included	Foreign language
Population (P)	Students Population not distinguished by gender/sex	Wrong population
Health as concept (C)	All other diagnosis/conditions than the included Missing or weak focus on women’s health	Wrong outcome
Measures and outcomes in working life (C)	Studies on interventions Medical interventions Pain relief Physical activity/lifestyle interventions Missing or weak focus on occupational life	Wrong outcome

**Table 3 ijerph-20-01080-t003:** Areas of women’s health with mapped conditions (*), sickness absence and work ability.

Areas of Women’s Health		Sickness Absence	Work Ability
Life stages of hormonal impact	39	25	30
PMS	7		
Dysmenorrhea	9		
Menstruation	4		
Endometriosis	13		
Menopause	19		
2. Pregnancy and reproduction	70	42	14
Pregnancy	56		
Breastfeeding	12		
Infertility	4		
Abortion	0		
3. Chronic and complex disorders	21	10	11
Chronic pain	14		
Headache/migraine	2		
Dizziness	0		
Fatigue/ME	4		
Fibromyalgia	7		
Other	3		
Total number of articles	130	77	55

* The described conditions overlap both in the three main areas of women’s health and in the outcome measurements of sickness absence and/or work ability.

## Data Availability

130 articles were included in this review. Included articles not cited in the text appear at the end of the reference list [122,123,124,125,126,127,128,129,130,131,132,133,134,135,136,137,138,139,140,141,142,143,144,145,146,147,148,149,150,151,152,153,154,155,156,157,158,159,160,161,162,163,164,165,166,167]. Other parts of the dataset are available from corresponding author on reasonable request.

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
