# Peer review of "Women’s Health and Working Life: A Scoping Review"

_ijerph, 2023, doi:10.3390/ijerph20021080_

Round 1
Reviewer 1 Report
The manuscript “Women’s Health and Working Life: A Scoping Review” is highly significant in the context of gender dimension of health.
Title
Suggested title: Women’s Health and Working Life in high-income countries: A Scoping Review
Reasons: In your review, all the countries belong to high-income categories except a few countries like Serbia, Bulgaria (World Bank classification https://datahelpdesk.worldbank.org/knowledgebase/articles/906519-world-bank-country-and-lending-groups).
Please remove the countries few non-high-income countries and highlight your review from high-income perspective.
Introduction:
The authors review the global evidence; however, the first paragraph of the Introduction is focused on Norway.
Make the Introduction more general. Gender-dimension of health status in high-income countries; working life patterns. The rationale behind the link between health and working life.
Need little medication in the flow of writing and content.
Methods
No issues
Results
Nicely organise.
If possible, describe the findings on the gender dimension of mental health and working life. How women faced mental health challenges due to their working life.
Discussion
Well written
Author Response
Dear Madam/Sir (Reviewer 1)
Thank you for reviewing our text and for helping us improving the manuscript. We have taken all comments and suggestions into consideration, and we believe the text has improved as a result. Please see our point-by-point response to the comments. The changes in the text will be visible with the use of “track changes” in the document.
Reviewers comments:
The manuscript “Women’s Health and Working Life: A Scoping Review” is highly significant in the context of gender dimension of health.
Title:
Suggested title: Women’s Health and Working Life in high-income countries: A Scoping Review
Reasons: In your review, all the countries belong to high-income categories except a few countries like Serbia, Bulgaria (World Bank classification https://datahelpdesk.worldbank.org/knowledgebase/articles/906519-world-bank-country-and-lending-groups). Please remove the countries few non-high-income countries and highlight your review from high-income perspective.
Introduction:
The authors review the global evidence; however, the first paragraph of the Introduction is focused on Norway.
Make the Introduction more general. Gender-dimension of health status in high-income countries; working life patterns. The rationale behind the link between health and working life.
Need little medication in the flow of writing and content.
Methods:
No issues
Results:
Nicely organise.
If possible, describe the findings on the gender dimension of mental health and working life. How women faced mental health challenges due to their working life.
Discussion:
Well written
Authors’ responses to reviewer 1
Title:
It is suggested to change the title to “Women’s Health and Working Life in high-income countries: A Scoping Review”, and to remove the countries which are not defined as high-income, like Serbia and Bulgaria, from the material. We will kindly ask you to reconsider this suggestion, based on the following arguments: The countries included in the review were selected by a desire to include countries with similarity to the Norwegian context regarding work life. Here Europe is especially relevant, since the labour marked is floating within the EU and through the marked agreements. Both Bulgaria and Serbia are fully European countries, Bulgaria is also a member of the EU and Serbia has applied for membership. In addition, other high-income countries were excluded from our material for the same reason. Based on this, we find the original selection more appropriate than to select studies due to the distinction of income of the countries. Hence, we also suggest keeping the original title.
Introduction:
We read the suggestion about not focusing on Norway in the introduction in connection with the suggested change towards a focus on high-income countries (above). Also, we like to keep Norway visible in the introduction for several reasons. Norway is a small country, but with a disproportionally large number of studies in this field (see figure 2 in the article, p.7). The authors are also Norwegian, and the first author will use this review article for her ongoing Ph.D.-work. Therefore, the example of Norway is valuable for the rest of the research project. We have now tried to make the introduction more general and international, as proposed, but with using Norway as an example, and hope that this solution can accommodate both interests. Changes appear on p. 1-2, line 28-52.
Results:
Mental health was defined within the exclusion criteria of the review, please see table 2, p. 5. (“All other diagnosis/conditions than the included”
We thank you again for reviewing our article and look forward to further process.
For the authors,
Marianne Gjellestad
Reviewer 2 Report
Dear Authors,
Please see my comments as follows:
Comments to Authors.
One Substantive comment that I wish to make is:
The issue of ' Legislation' related to Women's Working Life and OSH has received little attention in the paper ? The paper seems to take its lead from 'Employer accomodations' (e.g p9, line 270/ page 11 lines 343-345/ 347-349) - but does this not stem from societal requirement through Legislation in many countries ? . I feel that a modest paragraph outlining the connection of the study to ' legislation' would enhance the paper.
Minor typographical issues:
p10, line 310 meaning term 'avoid voiding' unclear to me as a reviewer.
p11 line 335 ' bothered by' alternative ' affected by'
p11 line 235 ' one can assume' alternative ' It is possible ...'
p11, line 372 'absence required by a doctor' alternative ' absence mandated by a doctor'.
Author Response
Dear Madam/Sir (Reviewer 2)
Thank you for reviewing our text and for helping us improving the manuscript. We have taken all comments and suggestions into consideration, and we believe the text has improved as a result. Please see our point-by-point response to the comments. The changes in the text will be visible with the use of “track changes” in the document.
Reviewers comments:
Dear Authors,
Please see my comments as follows:
One Substantive comment that I wish to make is:
The issue of ' Legislation' related to Women's Working Life and OSH has received little attention in the paper ? The paper seems to take its lead from 'Employer accomodations' (e.g p9, line 270/ page 11 lines 343-345/ 347-349) - but does this not stem from societal requirement through Legislation in many countries ? I feel that a modest paragraph outlining the connection of the study to ' legislation' would enhance the paper.
Minor typographical issues:
p10, line 310 meaning term 'avoid voiding' unclear to me as a reviewer.
p11 line 335 ' bothered by' alternative ' affected by'
p11 line 235 ' one can assume' alternative ' It is possible ...'
p11, line 372 'absence required by a doctor' alternative ' absence mandated by a doctor'.
Authors’ responses to reviewer 2
Comment on legislation
The comment on legislation is taken into the text at p.11, line 352-357. We thank you especially for this comment, as we agree that this is an important point that should be mentioned.
p10, line 310 meaning term 'avoid voiding'
The text says “delayed voiding”, describing how the nurses are hindered to use the facilities even if they need to, p.10, line 318-321.
p11 line 335 ' bothered by' alternative ' affected by'
The comment is taken into consideration and changes are made in the document, p. 11, line 344.
p11 line 235 ' one can assume' alternative ' It is possible ...'
The comment is taken into consideration and changes are made in the document, p. 11, line 367.
p11, line 372 'absence required by a doctor' alternative ' absence mandated by a doctor'.
The comment is taken into consideration and changes are made in the document, p. 11, line 386.
We thank you again for reviewing our article and look forward to further process.
For the authors,
Marianne Gjellestad
Reviewer 3 Report
The objective of this study was 12 to explore female physiology in a work-life context and to investigate possible associations be-13 tween women’s health, sickness absence and workability. A scoping review was conducted to de-14 develop a systematic overview of current research and to identify knowledge gaps. The search strategy 15 was developed through a population, concept and context (PCC) model, and three areas of women’s 16 health were identified for investigation in the work context.
Generally, this review is done at a high level and follows the JBI rules and recommendations.
Some points could be improved, and suggestions are as follow.
Line 82 - needs to refer to the protocol registration and link in what platform was done.
Line 84 methods: A preliminary search was done.? How did they arrive at creating the Boolean phrase?
Line 126 - was this research sting done for both databases? A table with the date of search, filters used and Boolean phrase and number of studies retrieved is standard and should be done.
No table that resumes the findings was made and would give the reader a good summary.
Author Response
Dear Madam/Sir (Reviewer 3)
Thank you for reviewing our text and for helping us improving the manuscript. We have taken all comments and suggestions into consideration, and we believe the text has improved as a result. Please see our point-by-point response to the comments. The changes in the text will be visible with the use of “track changes” in the document.
Reviewers comments:
Generally, this review is done at a high level and follows the JBI rules and recommendations.
Some points could be improved, and suggestions are as follow.
Line 82 - needs to refer to the protocol registration and link in what platform was done.
Line 84 methods: A preliminary search was done.? How did they arrive at creating the Boolean phrase?
Line 126 - was this research sting done for both databases? A table with the date of search, filters used and Boolean phrase and number of studies retrieved is standard and should be done.
No table that resumes the findings was made and would give the reader a good summary.
Authors’ responses to reviewer 3
Line 82 – protocol
The protocol was developed for internal use within the research team, it was not registered on any platform. It can of course be submitted to the reviewer as documentation, if desirable.
Line 84 methods: A preliminary search and Boolean phrase
The Boolean phrase was created by using the PCC-model (population, concept, context) and further by the definitions and selections done for the inclusion criteria. The preliminary search was done while developing the PCC, to check for width, appropriate timespan and number of articles, to make the search sufficiently comprehensive. The search was set up by using Emtree/Mesh browser.
Line 126 – research string, databases, table with the date of search, filters used Boolean phrase and number of studies
Table 1 (search string) at p. 4 is now updated with the number of studies retrieved from the search in Ovid (Embase/Medline). The search was done similarly in both databases. Numbers of retrieved articles from the different databases are specified in the flow chart of study selection, figure 1. , p. 6.
Table for summary
Findings are presented in table 3 and described in the text. Given the descriptive nature of a scoping review, we hope that table 3 gives an adequate summary of the findings, and that the readers and reviewers otherwise could retract a satisfactory summary from the text. However, we thank you for the comment and have now tried to clarify the explanation of the table (table 3, p. 7)
We thank you again for reviewing our article and look forward to further process.
For the authors,
Marianne Gjellestad